# Epidemiology and Clinical Presentation of West Nile Virus Infection in Horses in South Africa, 2016–2017

**DOI:** 10.3390/pathogens10010020

**Published:** 2020-12-30

**Authors:** Freude-Marié Bertram, Peter N. Thompson, Marietjie Venter

**Affiliations:** 1Department of Production Animal Studies, Faculty of Veterinary Science, University of Pretoria, Onderstepoort, Pretoria 0110, South Africa; bertramf@tut.ac.za (F.-M.B.); peter.thompson@up.ac.za (P.N.T.); 2Centre for Viral Zoonoses, Department of Medical Virology, Faculty of Health Sciences, University of Pretoria, Pretoria 0001, South Africa

**Keywords:** West Nile virus, horses, South Africa, epidemiology, emerging disease, encephalitis, neurotropic virus, zoonosis

## Abstract

Although West Nile virus (WNV) is endemic to South Africa (RSA), it has only become recognized as a significant cause of neurological disease in humans and horses locally in the past 2 decades, as it emerged globally. This article describes the epidemiological and clinical presentation of WNV in horses across RSA during 2016–2017. In total, 54 WNV-positive cases were identified by passive surveillance in horses with febrile and/or neurological signs at the Centre for Viral Zoonoses, University of Pretoria. They were followed up and compared to 120 randomly selected WNV-negative controls with the same case definition and during the same time period. Of the WNV-positive cases, 52% had fever, 92% displayed neurological signs, and 39% experienced mortality. Cases occurred mostly in WNV-unvaccinated horses <5 years old, during late summer and autumn after heavy rain, in the temperate to warm eastern parts of RSA. WNV-positive cases that had only neurological signs without fever were more likely to die. In the multivariable analysis, the odds of WNV infection were associated with season (late summer), higher altitude, more highly purebred animals, younger age, and failure to vaccinate against WNV. Vaccination is currently the most effective prophylactic measure to reduce WNV morbidity and mortality in horses.

## 1. Introduction

West Nile virus (WNV) is a neurotropic, zoonotic, vector-borne virus in the *Flaviviridae* family [1] and is a member of the Japanese Encephalitis virus sero-complex [2,3,4]. WNV was first identified in the West Nile District in Uganda in 1937 in a febrile human patient, since which periodic outbreaks were reported in Africa, the Middle East, and Europe [5]. Internationally, human WNV encephalitis was rarely encountered prior to early 1990s [6] but since then, outbreaks of increased severity, from new viral strains, likely of African origin, have occurred in parts of Europe and Asia. Since 1999, the Western Hemisphere was also affected, with substantial WNV disease incidence [7], and WNV has now become a significant globally re-emerging pathogen of importance in international trade [6]. WNV is regarded as the most geographically widely distributed arbovirus with increased incidence and severity of neurological disease in humans and horses as well as high mortality rates in birds in the Western Hemisphere [5], and it is one the leading causes of arboviral encephalitis globally [8,9].

WNV is maintained in nature by cyclic activity in numerous avian and mosquito species. African avian species are thought to be primary, reservoir hosts; they display no apparent signs of infection, which is presumably due to genetic resistance [6,10,11]. Mosquitoes may incidentally spread the virus to humans, horses, and other species, which then act as dead-end hosts due to the lower viraemia achieved in these species [5]. Approximately 20% of WNV infections in horses are symptomatic, with clinical signs ranging from fever to severe neurological signs (90%) and death [6,12]. Case mortality rates in unvaccinated horses range from 30 to 40% [13,14].

Two major lineages of WNV have been identified, lineage 1 and 2, and several less common geographic specific lineages [15,16,17,18]. Lineage 1 WNV is predominantly found in the Northern Hemisphere, including Europe, Northern Africa, the Middle East, parts of Asia, and Australia, where a closely related virus, Kunjin, clusters with lineage 1b. Lineage 1 was identified as the cause of deaths in birds, humans, and later horses in New York, U.S.A., in August 1999 [9], from where it spread across Northern America, Canada, and South America [5,15,19]. It is not known how the virus was introduced into the Americas, possibly through the legal or illegal importation of infected birds or the accidental importation of infected mosquitoes by aeroplane [6,9,20]. The WNV strain responsible for the initial North American outbreak was closely related to a WNV strain isolated from a dead goose in Israel during the previous year [19]. WNV-positive cases in horses and humans are reported annually in the U.S.A., with outbreaks differing in magnitude and geographic location, large outbreaks occurring every eight to ten years [21,22].

Lineage 2 WNV is most prevalent in Southern Africa and Madagascar, where it is endemic [2], and it emerged in 2006 in Central Europe from where it has spread to Greece, France, Italy, Germany, and causes frequent outbreaks of neurological infections in humans, horses, and birds [23,24]. Human WNV-positive cases were reported regularly in Europe since 1999, with increasing frequency of seasonal, regional outbreaks occurring in 2012 (935 cases), 2013 (785 cases), and 2018 (1670 cases and 124 deaths). The largest outbreak to date, in 2018, spread across 12 countries in southern and central Europe, and it was attributed to favourable climatic conditions, namely an early spring and very high temperatures during summer [21,25,26,27]. Although lineage 2 is predominantly associated with neurological infections of WNV in humans and animals in South Africa, a few lineage 1 strains have also been identified, suggesting that migratory birds may also import these strains to the region [28,29,30]. Passive surveillance of horses with febrile and neurological infections identified WNV cases across the county, particularly in Gauteng, KwaZulu-Natal, the Karoo and the Eastern Cape as well as the Western Cape [12]. Sequence identity amongst South African (RSA) lineage 2 strains indicated an exceptional constancy in the virus, strengthening the suspicion that local circulating foci of the WNV are being maintained in certain areas during the relatively mild, inland plateau winters [10,11]. However, migratory birds may have, on occasion, been the reservoir host responsible for the less common lineage 1 WNV infections [12,31].

Human cases of WNV fever have been consistently diagnosed in South Africa, with the largest outbreak in the Karoo in 1974 [32], followed by an outbreak in 1984 in Gauteng, after periods of unusually high rainfall and flooding in these areas [10]. Approximately 5–15 cases are reported annually by the National Institute for Communicable Diseases (NICD) [11,33]. During 2008–2009, WNV was detected in 3.5% of unsolved cases of human neurological disease in Gauteng provincial hospitals, indicating that WNV is underdiagnosed in human neurological cases [34,35]. This led to a study performed in 2011 to 2012 identifying South African veterinarians as a group with likely similar exposure risk as horses to WNV; 7.9% of veterinarians tested positive for antibodies against WNV, their distribution approximating that of WNV-positive cases detected in animals [36]. The World Organisation for Animal Health (OIE) reported West Nile virus infections in humans in South Africa from 2006 to 2018, with an average of four to ten cases annually. An increase in cases was reported in 2011 (52 cases) and 2012 (36 cases) and only one death was reported in 2014 [37].

The distribution of human outbreaks in South Africa was attributed to the ornithophilic *Culex univittatus* as the main mosquito vector (and to a lesser degree *Culex theileri, Culex pipiens*, and *Aedes caballus*) in the Highveld (central plateau) areas [10,38]. Given the right climatic conditions of heavy rains and higher than usual temperatures, *Cx. univittatus* has been responsible for significant WNV outbreaks in humans, despite having a low human feeding rate. Their eggs being very sensitive to desiccation, *Culex* spp. mosquitoes prefer temporary to semi-permanent rain flooded grassland, swamps, or other permanent water collections with emergent vegetation as breeding sites and survive dry winters as quiescent larvae and pupae or dormant adult females [38]. Both the mosquitoes’ gonotrophic period and the extrinsic incubation period of WNV in the insect vectors are very temperature dependent and can also be influenced by other environmental factors such as precipitation, hydrology, and humidity [5,39,40], which is why WNV disease tends to occur in late summer or autumn in the temperate, summer rainfall regions.

Research performed in 2000–2001 amongst South African Thoroughbred horses found that 11% of yearlings had already seroconverted against WNV, relative to sera collected approximately 12 months prior [25]. Of their dams, on these widely spread stud farms, 75% had also seroconverted, and yet no neurological clinical signs had been reported in any of these horses [41]. This is consistent with typical WNV occurrence worldwide, as most of the infected horses do not display overt clinical signs (approximately 80%), although viral encephalitis is seen in up to 90% of the symptomatic cases [6,12]. Systematic passive surveillance during 2008–2015 by the Centre for Viral Zoonoses (CVZ), University of Pretoria, confirmed a total of 79 clinical cases of WNV in horses in RSA, of which 91% displayed neuroinvasive disease, with a 34% case fatality rate [12].

Fever, particularly as the main syndrome, is an inconsistent finding in WNV-affected horses, especially when compared to other South African arboviruses such as Sindbis virus (SINV) and Middelburg virus (MIDV) [42]. It seems to be the only clinical sign in equine WNV infection that is not an exclusive reflection of central nervous system (CNS) pathology and may rather be attributed to the horse’s immune response to the viral infection. The cytolytic virus’ capacity to cause disease depends on its ability to survive in vivo, infect vital cells, and evade immune system recognition, inducing apoptosis in a diverse spectrum of tissues, including neurons [43]. WNV cases developing acute, progressive neuroinvasive disease [14] may show typical encephalomyelitis signs that may range from mild incoordination and weakness to severe ataxia, paresis or paralysis, recumbency, and death [27]. Neurological signs depend on the extent of CNS pathology and may include cranial nerve deficits, as summarised in Table 1 [1,5,6,44]. Up to 40% of recovered horses may show some form of persistent neurological deficit, either gait or behavioural abnormality, post recovery [13,14].

Currently, no specific treatment is available against WNV infection, but the American Association of Equine Practitioners (AAEP) guidelines recommend supportive treatment and nursing care aimed at reducing the CNS inflammation, preventing self-inflicted trauma, and providing nutrition and oral and intravenous fluid therapy as deemed necessary [5,45]. The control of the disease depends mainly on prophylactic vaccination to stimulate a protective immune response and mosquito management to avoid exposure to infected mosquitoes. Numerous studies have shown that protective immunity against WNV viraemia decreases both the severity of clinical signs as well as the mortality rate [1,5,6,13,45,46,47]. In RSA, an inactivated WNV vaccine is distributed by Zoetis (Duvaxyn), and a WNV recombinant canarypox virus vaccine is distributed by Merial/Boehringer Ingelheim (Proteq West Nile). These vaccines were licenced after epidemiological studies showed that WNV lineage 2 was associated with fatal neurological disease in horses [2], and a vaccine trial in mice showed that a lineage 1 vaccine cross-protected against lineage 2 WNV infection [48]. The WNV vaccine for horses was widely available in South Africa only as of 2015.

Passive surveillance for arboviruses such as WNV, Wesselsbron (WSLV; *Flaviviridae*), SINV (*Togaviridae*), and MIDV (*Togaviridae*), and Shuni virus (SHUV; *Peribunyaviridae*) has been routinely performed since 2008 for acute febrile and neurological disease in horses and other animals by the Zoonotic Arbo- and Respiratory Virus (ZARV) programme at the CVZ. During 2017, numbers of WNV-positive horses in RSA showed a remarkable increase from those diagnosed, on average, by the CVZ in 2008 to 2016.

The objective of this study was to investigate and describe the epidemiology and clinical case presentation of West Nile disease in horses in RSA from 2016 to 2017. Investigations included measuring the association of fever, acute neurological disease, and death, as well as certain predictor variables, with WNV infection. Predictors included animal demographic factors, vaccination status, environmental factors, and illness or stressful events within 4 weeks prior to sampling.

## 2. Results

A total of 54 WNV-positive cases (6 in 2016, 48 in 2017) were included in the analysis. The age range of WNV-positive horses was 4 months to 18 years (median 5 years). The highest proportion of WNV-positive cases was seen in younger horses (Figure 1), especially those less than 5 years old (*n* = 30, 55%).

In total, 15% (8/54) of the WNV-positive cases were co-infected with another virus (four with MIDV and four with equine encephalosis virus, EEV). Of the 120 randomly selected WNV-negative controls, 14 (12%) were MIDV-positive, 15 (12%) were EEV-positive, and 91 (76%) tested negative for all viruses on the ZARV programme testing panel. During the telephonic follow-up, it was determined that most of the controls were not definitively diagnosed. However, various confirmed diagnoses included herpes virus infection, tick-borne diseases (such as babesiosis, Karoo tick paralysis, and vestibular syndrome/facial paralysis from heavy auricular infestations), African horse sickness vaccine reactions, vertebral fracture, guttural pouch mycosis, brain tumours, and severe colic.

Neurological signs, with or without fever (rectal temperature >38.5 °C), were significantly more prevalent in WNV-positive cases than in WNV-negative controls (Table 2). Most of WNV-positive cases in 2016–2017 (48/54, 89%) displayed some neurological signs, of which 54% (26/48) had only neurological signs without fever. Approximately half of the WNV cases (28/54, 52%) had fever with or without neurological signs, fewer than the control group (76/120, 63%), although the difference was not significant (Table 3).

All types of paralysis (hindleg, foreleg, paresis, and total paralysis) as well as ataxia and tremors/muscle fasciculations were significantly associated with WNV infection (Table 3). Laminitic stance/sensitivity in the feet was noted in 9% (*n* = 5) of the cases and only one of the controls (*p* = 0.023). Recumbency tended to be more frequent in cases than controls (*p* = 0.091). Icterus, anorexia, and hyperreactivity/hyperaesthesia were not significantly associated with WNV infection.

Fatality proportions were similar for both cases (39%) and controls (36%) (Table 2). In both groups, subjects with only neurological signs had the highest fatality proportions (cases 14/26, 54% vs. controls 27/44, 61%), while those with only fever had the fewest fatalities. There was a tendency for WNV-positive cases with fever (with or without neurological signs) to be more likely to recover than those without fever (*p* = 0.057). Of the WNV cases that died, a larger proportion showed hindleg paralysis (9/21, 43%) and total paralysis (8/21, 38%) than foreleg paralysis (2/21, 10%) and tremors (2/21, 10%).

Of the 54 WNV-positive cases, 16 were euthanised due to a poor prognosis, with a median survival time of two days (range 0 to 469 days). Retained neurological signs after recovery were marginally more frequent in cases than in controls (Table 3), which were mainly related to ataxia or neurological instability. Three of these horses, an American Saddler and two Thoroughbred horses, were euthanised at 54–469 days after recovery, ranging in ages: 4 months, 6 years, and 18 years old. The 18-year-old Thoroughbred was also diagnosed with a cardiac tumour at euthanasia. In three other Thoroughbred horses (aged 2.5–3 years), performance was significantly affected by the retention of some degree of clinical signs resulting in early retirement from racing. One of them was sold as a pleasure hack even before racing; another was retired soon after attempting racing (following 8 months’ recuperation from WNV infection), and the third had a very unsuccessful racing career.

The breeds mostly represented in WNV-positive cases for 2016–2017 were Thoroughbreds (*n* = 26, 48%), Warmbloods (*n* = 9, 17%), and Arabian horses (*n* = 7, 13%), with the mixed (*n* = 3, 6%) and local breeds (Boerperd and Nooitgedachter) (*n* = 2, 4%) having significantly fewer cases than the purebred horses (*p* = 0.009). Neuroinvasive proportions tended to be similar (90–100%) amongst main breeds, but fatalities appeared to be fewer in Warmblood horses (22%); however, due to small numbers of individual breeds, statistical associations could not be made.

Spatial distribution of WNV cases (Figure 2) showed that most of the equine cases in 2016–2017 occurred in Gauteng (*n* = 19, 35%), KwaZulu-Natal (*n* = 14, 26%), and the Northern Cape (*n* = 11, 20%). The fewest cases were seen in the Western Cape (*n* = 5, 9%), Free State (*n* = 3, 6%), and North West provinces (*n* = 2, 4%), with no reported cases in Mpumalanga, Limpopo, and the Eastern Cape. The largest proportion of cases (*n* = 37, 69%) occurred at the 2nd and 3rd altitude quartiles (1057–1466 m).

In 2016, all six cases occurred during February–April. In 2017, the cases occurred during January–June, with a single case in early December in the winter rainfall region of the Western Cape. March was the month with the highest number of cases in both years (*n* = 28, 52%) and the greatest proportion of cases occurred during February–April (*n* = 46, 85%) (Figure 3).

Only 1/54 cases (2%) was reported to have been vaccinated against WNV in the 12 months before sample submission in 2016–2017, in comparison to 9/120 controls (8%). The WNV-positive case that had been vaccinated received an initial WNV vaccination 8 months and a booster vaccination 5 months prior to diagnosis, which were both administered by the owner. Of the 149 herd owners interviewed (cases and controls), only nine (6%) had vaccinated their horses in 2016–2017, whereas 30 (20%) had subsequently vaccinated in 2017–2018.

In the final multiple logistic regression model (Table 4), several variables were significantly associated with WNV infection. WNV cases were more likely to occur during March–April than any other time of the year (*p* = 0.007), and at an altitude of 1293-1466 m (*p* = 0.003). The odds of WNV diagnosis were the lowest in mixed and local breeds compared to intermediate and pure breeds, declined with increasing age (*p* = 0.041), and were lower in WNV-vaccinated horses (*p* = 0.047). Equine influenza virus (EIV) vaccination, although not significant (*p* = 0.145), was retained in the model as a confounder, as its removal resulted in substantial changes to the coefficients of the WNV vaccination and age variables. The Hosmer–Lemeshow test indicated the adequate fit of the final model (*p* = 0.094).

## 3. Discussion

The previous ZARV study of WNV in horses in RSA 2008–2015 [12] reported an average of 10 cases per year. The six WNV-positive cases in 2016 in the current study is consistent with that incidence rate; however, the 48 cases diagnosed in 2017 indicate a marked increase in case numbers. This could be due, in part, to increased awareness of WNV in RSA and sample submissions from suspected cases by owners and veterinarians. Increased awareness of WNV was created over the past few years by pharmaceutical companies’ product advertising as well as information disseminated through social media, veterinary congresses, scientific publications, and owner-targeted talks, many of which were facilitated by the ZARV programme, with feedback on WNV-positive cases. It is more likely that the sudden increase in WNV cases detected in RSA in 2017 was largely due to the environmental factors promoting the extensive breeding of WNV vectors, typical of the cyclical nature of these outbreaks, as seen in Europe and the U.S.A. [21]. Ecological and laboratory studies have determined the importance of environmental factors such as temperature, humidity, precipitation, and hydrology in WNV transmission dynamics, enabling mathematical modeling and the forecasting of potential outbreaks [40,49]. Overall, RSA experienced a severe drought in 2015–2016. During the early summer months of 2015/2016, there was very little rain and extremely high temperatures, followed by sudden high rainfall in the late summer months (February–April) of 2016. In particular, the eastern parts of the country had high rainfall in 2017, varying between 75% and 200% of the normal rainfall [50]. Higher rainfall following two dry seasons would have favoured the breeding of mosquitoes. Increased environmental temperatures may, up to a certain threshold, favour replication of the virus in the poikilothermic mosquito vectors, as well as decrease the subsequent length of the extrinsic incubation period and increase the efficiency of transmission of virus to susceptible hosts [40]. The seasonal variation in WNV cases was similar to those described in previous studies, occurring mainly in late summer and early autumn [12,51,52]. The slightly extended period of case distribution may have been attributed to the heavy rains, periodic flooding, and warmer than average temperatures of 2017, which followed the drought and extremely high environmental temperatures (El Niño conditions) of 2015–2016 [53]. In fact, 2017 was reported to be the fourth warmest year in South Africa since 1951, with 2015 and 2016 the two warmest recorded years [53].

The spatial distribution of the 54 WNV-positive cases in horses as detected by the ZARV programme, CVZ, in 2016–2017, followed a similar pattern to those previously detected in earlier years by the ZARV [12]. The largest proportion of cases was detected on the Highveld, mainly in the warm to temperate, summer rainfall zones in the eastern parts of RSA, which was likely due to ideal vector breeding conditions [10,38]. Of the 586 equine test submissions to ZARV in 2016–2017, Gauteng had the highest number of submissions (*n*= 264, 45%) as well as the highest number of WNV-positive cases (*n* = 19, 35%). Fewer cases were seen in the Western Cape and surrounding areas, which was most likely due to the Mediterranean climate and winter rainfall patterns being less favourable to vector development and not due to a lack of valuable horses or to the underreporting of cases. Only a small proportion of the samples from the Western Cape tested positive for WNV (*n* = 5/109, 5%), despite the submission of a large number of samples from that region (*n* = 109, 19% of total sample submissions). Most of the WNV-positive cases that were diagnosed in Thoroughbreds were from stud farms located in KwaZulu-Natal (*n* = 10), Western Cape (*n* = 5) and the Northern Cape (*n* = 4). According to a stud farm survey done by the Thoroughbred Breeders’ Association early in 2017 [54], 65% of the Thoroughbred stud horses were located in the Western Cape province, 22% were located in KwaZulu-Natal, and 1% was located in Gauteng. Of those specified on participating stud farms, 52% were classified as youngstock under 3 years old and 45% were classified as adult breeding stock. Only 4% were reported to be non-breeding adult Thoroughbred horses at stud with a negligible number of horses of other breeds (<1%).

Most of the WNV cases were diagnosed using immunoglobulin M (IgM) enzyme-linked immunosorbent assay (ELISA), rather than rtRT-PCR. This is consistent with international findings and is due to the short-lived viraemia [6]. Recent increased exposure resulting in persistent IgM levels in convalescent cases, which may then be incorrectly diagnosed as acute cases, were unlikely, as the WNV-positive cases in this study fit the typical clinical description of acute WNV-positive cases. The total co-infection rate for 2016–2017 approached the 18% level previously reported [12]; however, an increased co-infection rate with both MIDV and EEV was observed during 2016–2017 when compared to previous years [17]. Larger outbreaks of these viruses were detected in 2017, which may be the reason for this, but this will be reported elsewhere.

Two thirds of the WNV-positive deaths were due to elective euthanasia, which was presumably due to either a grave prognosis or economic constraints affecting treatment. The extremely short median survival time until elective euthanasia was likely attributable to the rapid onset of severe clinical signs. Case fatality and neuroinvasive proportions in 2016–2017 were similar to those previously reported both in RSA [12] and internationally [13,14,55].

The most prevalent clinical signs displayed by cases during 2016–2017 were very similar to those described for equine WNV-positive patients in general, consisting mainly of various neurological signs with or without fever [5,6,12,27,55,56]. As with the previous ZARV study [12], neurological signs were present in most WNV-positive cases, and all neurological clinical signs were significantly associated with WNV infection (*p* = 0.001). A consistently high percentage of neuroinvasion was seen among all age groups that tested WNV-positive. In both WNV cases and controls, subjects with only neurological signs had the highest odds of fatality, which is likely due to the severity of pathology in the CNS. It was interesting that the fatal WNV cases had higher proportions of hindleg and total paralysis than foreleg paralysis and tremors or muscle fasciculations, relating to the degree and location of spinal cord pathology. A significant, consistent decrease both in absolute WNV case numbers and in odds of WNV infection relative to other causes of neurological and febrile disease, was seen with increasing age (*p* = 0.041), which was possibly due to an increased immunological resistance resulting from repeated low-grade exposure to WNV [5,12,13]. This is in contrast to human WNV cases in whom clinical signs are more apparent and severe in very young and very old patients, with neurological signs especially in the elderly [7,43,57]. The literature differs in opinion regarding whether the age of horses influences the case fatality rate or whether geriatric horses are more or less susceptible [13,58], but it is generally agreed that horses younger than 5 years are more susceptible to clinical disease. In rodents, increased viraemia correlated with earlier neuroinvasion and increased WNV burdens in the CNS [43]. Studies using South African lineage 2 strains showed that neuroinvasive LD50 levels for most strains were comparable to those of lineage 1 strains in the USA, although some strains were more neuroinvasive than others [59].

Fever, when present, was presumably due to a systemic inflammatory reaction to the WNV infection [43]. Approximately half of the WNV-positive cases in 2016–2017 had fever compared to 35% in 2008–2015 [12]. As in human patients [7], uncomplicated equine WNV cases usually recovered fully. The tendency for WNV cases with fever, with or without neurological signs, to be more likely to recover than those without fever was surprising. This can also be seen in another study that detailed the clinical presentation of lineage 2 WNV cases in horses in Austria in 2016 to 2018 [27]. Studies have shown that numerous attributes of the innate and adaptive immunity, particularly T-cell mediated immunity, are required to successfully counteract the viraemia and mitigate pathogenesis in the CNS [43]. The presence of fever may be an indication of the horse’s ability to mount an effective general immune response to the initial viraemia. Therefore, the presence of certain clinical signs, such as fever, could potentially serve as a prognostic indicator for recovery.

Laminitic stance/sensitivity in feet was an interesting clinical sign noted in a small but statistically significant number of cases (*p* = 0.023). It is not generally described in the literature as a sign of equine WNV infection, although one study mentioned “reluctance to move” as a clinical sign [1] and another mentioned “hindlimb lameness/monoplegia” [27]. Some human WNV patients may experience severe pain in their limbs just before or during the onset of weakness [7], suggesting a neuropathic cause for the perceived pedal sensitivity in some of the equine patients. All five of these horses also displayed neurological signs, and two were eventually euthanised. Retained clinical signs such as gait or behavioural abnormality after recovery were seen in a small, marginally significant number of cases (*p* = 0.053), although there were much fewer than reported elsewhere [13,14]. It is possible that there would have been a higher proportion of retained clinical signs in South African horses if fewer economic constraints and more awareness of the disease had allowed a longer treatment period.

The various mixed breeds and the South African breeds, Boerperd and Nooitgedachter, were associated with a significantly lower odds of WNV infection compared to purebred and intermediate hybrid vigour groups (*p* = 0.009). These horses may show greater immunological resistance to endemic diseases due either to hybrid vigour or to some form of genetic adaptation, as seen in humans with certain gene mutations [60] or asymptomatic WNV infection in birds from endemic countries such as RSA [10]. Accurate data are not available, but a large proportion of the general South African equine population is likely comprised of non-purebred indigenous/local or mixed breeds; they were also well represented in the randomly selected controls, indicating that they were not overlooked due to being of less economic value.

Stabling at night, sex of horse, vaccination against African horse sickness, and being generally highly stressed in the four to six weeks before clinical symptoms were detected were not found to be significantly associated with WNV infection in the univariate analysis. It is important to note that a large proportion of the subjects (both cases and controls) had, according to the owners, experienced high levels of stress, particularly due to long-distance travelling, in the 4–6 weeks before sample submission. Research in rodents showed that increased stress levels promoted immunosuppression, increased WNV replication in vivo, and increased neuroinvasion causing encephalitis and death [20,61]. Considering that all horses in this study, including controls, were diseased, the effect of stress on WNV infection could not be evaluated; however, this suggests that certain stressors may play a role in equine disease development in general.

Being vaccinated against EIV may have been acting as a proxy for other unmeasured variable(s) associated with WNV infection risk, leading to its being retained in the final model as a confounder. EIV vaccination is compulsory for competitive sport horses, required for participation in events organised by the South African Equestrian Federation (SAEF) and other individual discipline associations such as the National Horse Racing Authority (NHRA). Competitive horses vaccinated against EIV would typically be more likely to travel than horses that are not vaccinated against EIV. Thus, it is possible that the stress of competing, travelling, or exposure of an immunologically naïve horse to WNV or to different regional strains of WNV, or another associated factor, may predispose a horse to contracting WNV disease.

Based on the follow-up of owners of affected horses, a much higher proportion vaccinated their herds after sample submission in 2017 to 2018 compared to 2016 to 2017. However, this number only reflects the study subjects from the CVZ database and is not necessarily an indication of increased vaccination proportion countrywide. Although vaccine sales data are not available, personal communication with one of the two pharmaceutical companies currently selling a WNV vaccine in RSA has confirmed that sales of the WNV vaccine increased during 2018 to 2019. This was noted particularly after the period of telephonic follow-up interviews with the owners and veterinarians, during which WNV infection and available vaccinations were explained. Awareness campaigns in recent years have also created public recognition of the potential for WNV to cause severe neurological disease and death in horses and humans in RSA and of the need for vaccination, which should result in decreased numbers of WNV-positive cases in horses in South Africa. However, due to economic constraints, or perhaps ignorance regarding immunology, one may expect many South African horse owners to neglect regular vaccination during years when few cases occur, as is the case with other non-government regulated vaccines. This may contribute to a decline in immunity, especially in younger, immunologically naïve horses, leading to increased WNV case numbers during outbreaks. There is also uncertainty amongst horse owners regarding the duration of natural immunity post-infection, creating differing opinions regarding whether horses should be vaccinated.

## 4. Materials and Methods

Ethical approval: The study protocol (V080-18), as well as previous testing of CVZ samples (H01216), was approved by the University of Pretoria Animal Ethics Committee and the Faculty of Veterinary Science Research Committee. Department of Agriculture, Land Reform and Rural Development (DALRRD) approval had been obtained for testing of CVZ samples under Section 20 approval. Annual reports of the cases detected by the CVZ were submitted to the DALRRD. Veterinarians and owners involved were informed of the purpose of the questionnaire and assented either verbally or electronically, as well as by written permission on the test requisition form, to the information being used for research purposes.

Study design: A case-control study was conducted using submissions to the ZARV programme during 2016 to 2017. Criteria for submission to the study were one or more of the following: fever (more than 38.5 °C) and/or acute neurological disease, and/or death. Neurological disease was characterised as horses displaying one or more of the following clinical signs: ataxia, blindness, facial paralysis, hyperreactivity or hyperaesthesia, incoordination, nystagmus, paresis, partial or complete paralysis, recumbency, seizures, tremors and muscle fasciculations, tongue paralysis and/or weakness, lip twitching, head tilting, and/or dysphagia.

Cases were defined as the horses in the ZARV database that tested positive for WNV infection on real-time reverse transcriptase polymerase chain reaction (rtRT-PCR) using the method published in [62], immunoglobulin M enzyme-linked immunosorbent assay (IgM ELISA) (IDEXX IgM kit, Hoofddorp, The Netherlands), or both. IgM-positive cases were confirmed by serum neutralisation assays in the BSL3 laboratory at the CVZ. All 54 available WNV-positive cases were used, of which 6 horses were diagnosed WNV-positive in 2016 and 48 were diagnosed WNV-positive in 2017. Most of the WNV-positive cases were diagnosed using IgM ELISA (47/54, 87%) of which two cases (4%) tested positive on both real-time reverse transcription polymerase chain reaction (rtRT-PCR) and ELISA. Only 17% (9/54) of the cases tested positive for WNV on rtRT-PCR. For each case, at least two WNV-negative control horses were randomly selected from the same population of ZARV sample submissions. These 120 control horses also complied with one or more of the three inclusion criteria. Any incomplete information on the sample requisition forms submitted for the subjects was obtained from the veterinarians who sent the samples or the owners of the horses. Interviews with owners also assessed general awareness of WNV, common clinical signs, potential pathological course, occurrence and distribution in RSA, and availability and use of vaccines. Additional information was obtained during telephonic interviews regarding the general vaccination status of the horses, potential stressors in the four to six weeks prior to displaying clinical signs (such as long-distance travelling, change of ownership and management system, recent illness or injury, weaning or recent African horse sickness vaccination), housing at night, and outcome of the disease.

Since some of the owners had multiple submissions to the dataset (for instance at a stud), responses regarding WNV vaccination were also grouped into herds based on location rather than individual horses. A total of 149 herds were interviewed (cases and controls), of which 47 herds contributed to the WNV-positive cases.

Horses were also grouped according to age; distribution of the WNV-positive cases was as follows: 56% (*n* = 30) of cases were juveniles less than 5 years old, 39% were adult horses between the ages of 5 and 15 years old (*n* = 21), and 5% were considered geriatric, being older than 15 years (*n* = 3).

Data Analysis: Comparisons between years were done using all available data from the 583 CVZ sample submissions of horses during 2016 and 2017. Analysis of the case-control study was done using the 174 subjects on which complete data had been obtained. Variables for analysis were divided into 2 groups: the exposure or potential risk factors plausibly associated with the likelihood of a horse being WNV-positive were used to develop a multiple logistic regression model. Secondly, the recorded clinical signs and outcomes (partial/full recovery, death/euthanasia) were described and correlated with the presence or absence of WNV infection by means of univariate analysis.

Univariate analysis of risk factors was performed using cross-tabulation and two-tailed Fisher’s exact test. For continuous variables, the assumption of linearity was assessed by plotting the Pearson and Deviance residuals against the value of the predictor in a simple logistic regression model and evaluating the linearity of the resulting pattern. The predictors were also categorised into quartiles, and the quartile midpoints were plotted against their estimated log odds to evaluate linearity. Elevation above sea-level (altitude) of the locations of WNV-positive cases was categorised into quartiles due to being non-linear. Some categorical variables were recoded to increase statistical power by combining categories with few observations.

Multivariable analysis was done using multiple logistic regression. For the initial logistic regression model, all univariate risk factor variables with the likelihood ratio test (LRT) *p* < 0.2 were included, and the least statistically significant variables were eliminated using a backward stepwise procedure. Finally, all independent variables were re-included one by one and retained if significant, or if inclusion resulted in substantial (>20%) changes in the coefficients for other variables in the model. Goodness of fit was evaluated using the Hosmer–Lemeshow goodness-of-fit statistic. Analysis was done using Stata 15 (StataCorp, College Station, TX, USA) and NCSS 2007 (NCSS, Kaysville, UT, USA). Significance was set at *p* < 0.05.

## 5. Conclusions

Increased equine WNV-positive case numbers in RSA in 2017 were largely attributed to environmental factors favouring the breeding habits of the vector. The largest proportion of cases during 2016–2017 was reported in the temperate to warm, eastern inland RSA plateau, at intermediate elevation above sea level, during March–April. The WNV-associated case fatality rate and neuroinvasive disease proportions from 2016 to 2017 were consistent with those reported in previous local and international studies. Most of the cases displayed neurological signs, which were significantly associated with WNV infection, and approximately half of the cases had fever. Fever was marginally associated with recovery from WNV and may potentially be used as a prognostic indicator. Vaccination against WNV was significantly protective, and the risk of developing clinical WNV significantly decreased with increasing age, which was likely due to increased immunity from repeated long-term, low grade field exposure.

Therefore, it is advisable that owners with competitive horses or those younger than two to five years old, especially the highly purebred breeds (such as Thoroughbreds, Warmbloods, and Arabians) residing in the eastern temperate to warm parts of RSA with high summer rainfall, or travelling between provinces, should practice routine, complete vaccination against WNV. These vaccines should be given annually during spring, in order to decrease morbidity and mortality by timeously increasing immunological resistance against WNV.

## Figures and Tables

**Figure 1 pathogens-10-00020-f001:**
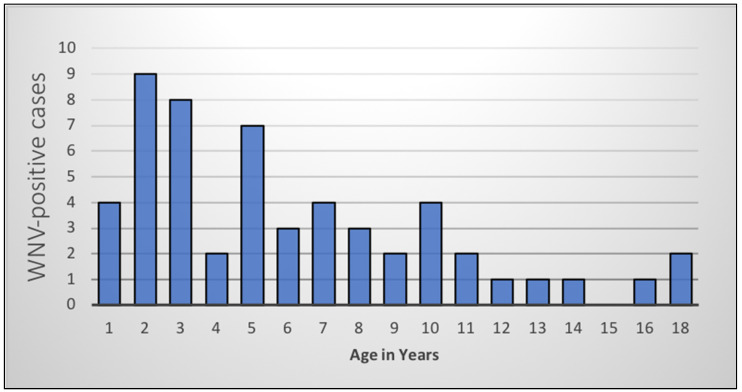
West Nile virus-positive cases in horses in South Africa by age in years, 2016–2017.

**Figure 2 pathogens-10-00020-f002:**
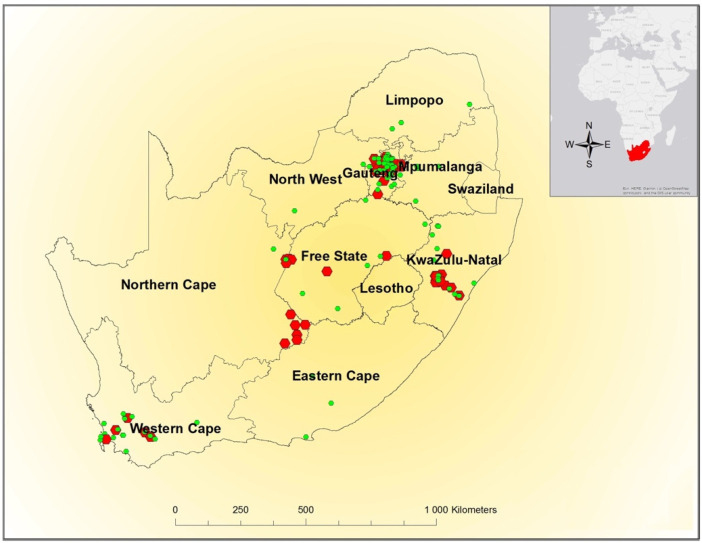
Distribution of West Nile virus-positive cases (red) and WNV negative controls (green) in horses used in the study, by province, 2016–2017. Each marker may represent one or more cases at the same location.

**Figure 3 pathogens-10-00020-f003:**
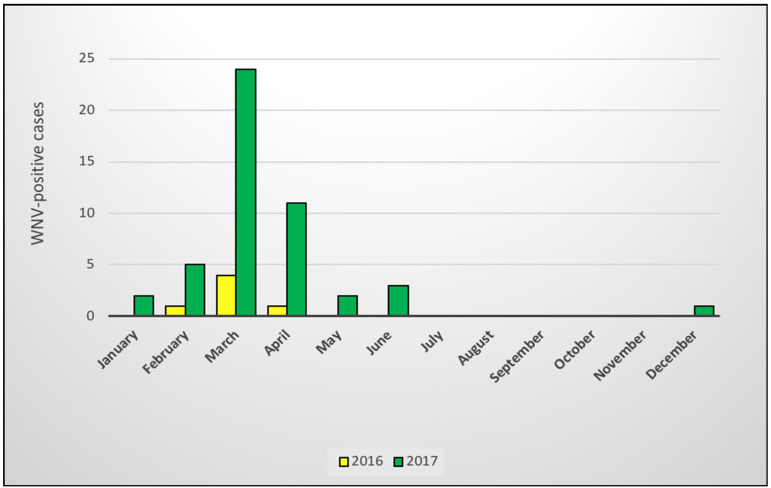
Distribution of WNV-positive cases in horses in South Africa by month in which the initial signs were displayed, 2016–2017. The yellow bars show the 2016 cases and green bars show the 2017 cases.

**Table 1 pathogens-10-00020-t001:** Typical neurological signs of West Nile virus (WNV) disease in horses as related to damage to three areas of the central nervous system.

CNS Location of WNV Pathology	Typical Neurological Signs Associated with Specific Area of Pathology
Spinal cord pathology	Weakness, ataxia, reluctance to move, paresis, or paralysis affecting one or more limbs, skin or muscle fasciculations, muscle tremors and muscle rigidity. Paralysis of hindlimbs (“dog-sitting”); progressive paralysis of all four limbs usually ending in recumbency.
Brain pathology *	Ataxia, dysmetria, hyperaesthesia, and abnormal mentation (ranging from somnolence and depression to agitation and hyperexcitability, even aggression)
Cranial nerve deficits	Facial nerve paralysis (VII) including droopy lip, muzzle deviation, or lip twitching. Tongue weakness or paresis (XII), head shaking, head tilt (VIII), and dysphagia (IX, X). Fine tremors of the face (VII) and neck muscles (XI).

* Pathological changes in medulla oblongata, pons, thalamus, reticular formation, cerebellum and brain cortex.

**Table 2 pathogens-10-00020-t002:** Disease outcome of West Nile virus-positive cases and WNV-negative controls, by main syndrome, amongst horses detected by passive surveillance of neurological and febrile cases in South Africa, 2016–2017.

Disease Outcome		Fever Main Syndrome	Neuro Main Syndrome	Fever and Neuro Main Syndrome	Total
WNV-positive cases	Deaths	1	14	6	21 (39%)
	Recovered	5	12	16	33 (61%)
	Total	6 (11%)	26 (48%)	22 (41%)	54
WNV-negative controls	Deaths	7	27	9	43 (36%)
	Recovered	36	17	24	77 (64%)
	Total	43 (36%)	44 (37%)	33 (28%)	120

**Table 3 pathogens-10-00020-t003:** The most important clinical signs in the 54 West Nile virus-positive cases and 120 WNV-negative controls, amongst horses detected by passive surveillance of neurological and febrile cases in South Africa, 2016–2017.

Clinical Signs	WNV-Positive		WNV-Negative		Odds Ratio	95% CI	*p*-Value *
	*n*	*%*	*n*	*%*			
Died	21	39%	43	36%	1.1	0.6, 2.3	0.824
Euthanised	16	30%	25	21%	1.6	0.7, 3.5	0.284
Retained signs post-recovery	6	11%	3	3%	4.5	1.0, 31.0	0.053
Neurological signs	48	89%	77	64%	4.2	1.7, 13.7	0.001
Ataxia	40	74%	59	49%	2.9	1.4, 6.5	0.003
Fever	28	52%	76	63%	0.6	0.3, 1.3	0.208
Hindleg paralysis	19	35%	22	18%	2.4	1.1, 5.3	0.028
Recumbency	18	33%	24	20%	2.0	0.9, 4.4	0.091
Paresis	16	30%	18	15%	2.4	1.0, 5.5	0.045
Paralysis	15	28%	14	12%	2.9	1.2, 7.1	0.018
Icterus	11	20%	28	23%	0.9	0.3, 1.9	0.823
Tremors, fasciculations	10	19%	8	7%	3.1	1.1, 9.9	0.041
Foreleg paralysis	9	17%	4	3%	5.4	1.5, 26.8	0.008
Anorexia	8	15%	28	23%	0.6	0.2, 1.4	0.278
Laminitic stance	5	9%	1	1%	8.9	1.3, -	0.023
Hyperreactive/Hyperaesthetic	5	9%	3	3%	3.7	0.7, 26.4	0.124

* Two-tailed Fisher’s exact test; significance set at *p* < 0.05.

**Table 4 pathogens-10-00020-t004:** Final logistic regression model of factors associated with WNV-infection in South African horses with acute febrile or neurological disease detected by the Zoonotic Arbo- and Respiratory Virus (ZARV) programme at the Centre for Viral Zoonoses (CVZ), 2016–2017.

Variable	Level	Odds Ratio	95% CI	*p*-Value
*Month*	January–February	5.4	0.6, 49.9	0.134
	March–April	18.0	2.2, 149.5	0.007
	May–June	4.2	0.4, 44.9	0.241
	July–December	1 *	–	–
*Altitude*	16–1056 m	1 *	–	–
	1057–1292 m	1.2	0.4, 3.9	0.807
	1293–1466 m	6.0	1.9, 19.1	0.003
	1467–1784 m	1.2	0.3, 4.3	0.764
*WNV vaccinated*	Yes vs. no	0.1	0.0, 1.0	0.047
*Age in years*	Continuous	0.9	0.9, 1.0	0.041
*Breed*	Highly Purebred	3.0	0.9, 9.7	0.068
	Intermediate Hybrid vigour	4.9	1.3, 18.2	0.019
	Mixed and Local	1 *	–	–
*Equine influenza virus vaccinated*	Yes vs. no	2.1	0.8, 5.6	0.153

* Reference level.

## Data Availability

The data were collected as part of the research and surveillance program by the ZARV programme in the Centre for Viral Zoonoses, University of Pretoria. The data is not publicly available to protect the identity of the owners. The minimal dataset that supports the central findings of a published study may be obtained from the corresponding author, M.V. under agreement with the University of Pretoria.

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
