# Peer review of "Epidemiology and Clinical Presentation of West Nile Virus Infection in Horses in South Africa, 2016–2017"

_pathogens, 2020, doi:10.3390/pathogens10010020_

Round 1
Reviewer 1 Report
The manuscript by Bertram et al., (Manuscript ID: pathogens-1039093) entitled " Epidemiology and clinical presentation of West Nile virus infection inhorses in South Africa, 2016–2017" presents a clinical and epidemiological study carried out on horses with clinical neurological signs and or fever that tested positive for WNV, compared with horses presenting similar signs but WNV-negative. The study associates the increase in WNV-positive cases in the studied period with environmental factors.
In general the work is well conducted, although there are some concerns to be resolved. An important matter is that bibliography needs to be deeply updated. There is only a few references with less than 5 years, and sentences like lines 73-74 (reference 11) are quite old and should be revised, since data are almost 20 years old. This also happens in lines 59-60, since outbreaks in Europe have increased in the past few years.
Other concern that seems a little surprising is that Table 2 shows similar mortality rates among WNV-positive and negative cases. Which is the possible explanation of these rates, since WNV-negative controls tested also negative to other viruses such as EEV and MIDV in a 76%?
Although the main conclusion for the increase in WNV-positive cases is the link with the environmental factors, it would be worthy to discuss if there could be other possible factors, such as the increase in the implementation of surveillance programmes or a better detection of the disease.
Other minor points:
Data of the number of cases along the year would be easy to follow if presented in a figure.
Line 31. It is more accurate to write Flaviviridae family instead of family Flaviviridae. Please change it
Author Response
Comments from Reviewer 1
Comment 1: An important matter is that bibliography needs to be deeply updated. There is only a few references with less than 5 years, and sentences like lines 73-74 (reference 11) are quite old and should be revised, since data are almost 20 years old. This also happens in lines 59-60, since outbreaks in Europe have increased in the past few years.
Response: Thank you for your insightful suggestion. We have, accordingly, modified the references to emphasize these points. Reported human West Nile virus cases in South African have been inserted from the WAHIS interface (refer: lines 81, 86 - 89 references 33, 37). The NICD was unfortunately not able to provide us with more detailed data at present.
The World Organisation for Animal Health (OIE) reported West Nile virus infections in humans in South Africa from 2006 to 2018, with an average of 4 to 10 cases annually. An increase in cases was reported in 2011 (52 cases) and 2012 (36 cases) with only 1 reported death, in 2014 [37].
More recent references were added regarding the European and USA outbreaks (refer: lines 58 – 60, 63 - 68, references 21, 22, 25 - 27).
WNV-positive cases in horses and humans are reported annually in the U.S.A., with outbreaks differing in magnitude and geographic location, large outbreaks occurring every eight to ten years [21, 22]
Lineage 2 WNV is most prevalent in Southern Africa and Madagascar, where it is endemic [2], and emerged in 2006 in central Europe from where it has spread to Greece, France, Italy, Germany and causes frequent outbreaks of neurological infections in humans, horses and birds [23, 24]. Human WNV-positive cases were reported regularly in Europe since 1999, with increasing frequency of seasonal, regional outbreaks occurring in 2012 (935 cases), 2013 (785 cases) and 2018 (1670 cases and 124 deaths). The largest outbreak to date, in 2018, spread across 12 countries in southern and central Europe, and was attributed to favourable climatic conditions, namely an early Spring and very high temperatures during Summer [21, 25-27]
Comment 2: Other concern that seems a little surprising is that Table 2 shows similar mortality rates among WNV-positive and negative cases. Which is the possible explanation of these rates, since WNV-negative controls tested also negative to other viruses such as EEV and MIDV in a 76%?
Response: Thank you for pointing this out, this highlights an interesting aspect which should have been addressed. Therefore, possible explanations for the mortality rates in these WNV-negative controls was added (refer: lines 157 - 161).
“During the telephonic follow-up, it was determined that most of the controls were not definitively diagnosed. Various confirmed diagnoses, however, included herpes virus infection, tick borne diseases (such as babesiosis, Karoo tick paralysis and vestibular syndrome/ facial paralysis from heavy auricular infestations), African horse sickness vaccine reactions, vertebral fracture, guttural pouch mycosis, brain tumours and severe colic.”
Comment 3: Although the main conclusion for the increase in WNV-positive cases is the link with the environmental factors, it would be worthy to discuss if there could be other possible factors, such as the increase in the implementation of surveillance programmes or a better detection of the disease.
Response: We agree with this and have incorporated your suggestion in the manuscript. Some of these reasons were already discussed but have consequently been elaborated as suggested (refer: lines 243 - 251).
“Increased awareness of WNV was created over the past few years by pharmaceutical companies’ product advertising as well as information disseminated through social media, veterinary congresses, scientific publications and owner targeted talks, many of which were facilitated by the ZARV program, with feedback on WNV-positive cases. It is more likely that the sudden increase in WNV cases detected in RSA in 2017 was largely due to the environmental factors promoting extensive breeding of WNV vectors, typical of the cyclical nature of these outbreaks, as seen in Europe and the U.S.A. [21]. Ecological and laboratory studies have determined the importance of environmental factors such as temperature, humidity, precipitation and hydrology in WNV transmission dynamics, enabling mathematical modelling and forecasting of potential outbreaks [40, 49]”
Comment 4: Data of the number of cases along the year would be easy to follow if presented in a figure.
Response: Thank you for this suggestion. We have inserted a graph to illustrate this better (refer: Figure 3, line 217 – 220 )
Comment 5: Line 31. It is more accurate to write Flaviviridae family instead of family Flaviviridae. Please change it
Response: Thank you for pointing out the error, it has been corrected (refer: line 31)
Reviewer 2 Report
Dear authors,
Please find bellow a very short recommendation and suggestion list to consider. The topic you approached is very interesting and valuable. It also opens ways for further analysis and study, especially for the given geographical area of South Africa and possibly different disease patterns when compared to other continents and countries.
Introduction
L47: citation needed for the 90% of severe neurological signs’ occurrence (probably the same as in L98?)
L135: consider to change’ 2016-2017’ to ‘from 2016 to 2017’
Results
L140-150: the number of horses in the study and the diagnostic tests belong to the ‘Material and methods’ section
L167: please state the statistical significance value set for p (in the Materials and methods section)
L179-181: being more precise would add value
L199-202: can be considered description of the study conditions, it would belong to the ‘Materials and method’ section
Discussion
L283-287: Please document and mention studies establishing increased severity of neurological manifestations of WN fever in older horses compare to younger ones (also relative to immunity). You results showing a different pattern, it would worth a more in-depth look
L297-299: citation needed
L302: please insert the value of p in brackets after ‘statistically significant’ here and in similar instances (L308, L313 and so on)
L342: It might be interesting to state one out of how many pharmaceutical companies selling the WNV vaccine in the RSA?
L342-346: According to the experience of most European countries, the demand for the WNV vaccine increases in those years with increased prevalence of the clinical disease (i.e. 2018 in the closest past). Then, because of the high numbers of asymptomatic individuals that gain long term acquired natural immunity and also the contribution of vaccination protocols, the following years record drastically less WN fever outbreaks, mainly until the increase of the numbers of naïve horses (new generations). In these ‘silent’ years horse owners’ awareness for vaccination change which usually leads to marked fluctuations of the WNV vaccine market, posing possible challenges for the vaccine producers. It might be of interest for the international reader community to have a South African opinion about this.
L345-346: please consider to rephrase ‘should have a positive effect on the numbers of WNV positive cases in horses in South Africa.’ (avoid repetition of ‘positive’; a positive effect on the positive cases involuntarily suggests an increase, which is not the case here; an addition on mentioning immunity development through both symptomatic and symptomless natural disease…)
Materials and methods
L139: Please describe a bit more the materials and methodology of the study. For example, more details of the animals if possible (numbers of horses in each age group, use of horses, also with emphasis on their housing for possible risk factors for contracting the disease, approximate numbers of horses in the same facility for case prevalence inside an outbreak) could add benefit.
Author Response
Comments from Reviewer 2
Comment 1: L47: citation needed for the 90% of severe neurological signs’ occurrence (probably the same as in L98?)
Response: Thank you for pointing this out, the references are indeed the same and were inserted as suggested (refer line 48, reference 12).
Comment 2: L135: consider to change’ 2016-2017’ to ‘from 2016 to 2017’
Response: Thank you for the suggestion, it has been corrected (refer line 145). It has also been changed elsewhere in the manuscript, to facilitate uniformity.
Comment 3: L140-150: the number of horses in the study and the diagnostic tests belong to the ‘Material and methods’ section
Response: Thank you for this suggestion. The diagnostic tests have been moved to the materials and methods section (refer lines 407 – 411)
Comment 4: L167: please state the statistical significance value set for p (in the Materials and methods section)
Response: Thank you for the suggestion; this has been included in a footnote to the table (refer line 178).Significance set at p<0.05
Comment 5: L179-181: being more precise would add value
Response: Thank you for indicating the need for more precision in this section, due to the length of the article a lot of the detail had to be summarized. As requested, it has been reinserted into the text and we trust that it will read more fluently now. (refer lines 189 – 195)
“Of the 54 WNV-positive cases, 16 were euthanised due to a poor prognosis, with a median survival time of two days (range 0 to 469 days). Retained neurological signs after recovery were marginally more frequent in cases than in controls (Table 3), mainly related to ataxia or neurological instability. Three of these horses, an American Saddler and two Thoroughbred horses, were euthanised at 54–469 days after recovery, ranging in ages: 4 months, 6 years and 18 years old. The 18 year-old Thoroughbred was also diagnosed with a cardiac tumour at euthanasia. In three other Thoroughbred horses (aged 2.5–3 years) performance was significantly affected by the retention of some degree of clinical signs resulting in early retirement from racing. One of them was sold as a pleasure hack even before racing, another was retired soon after attempting racing (following 8 months’ recuperation from WNV infection) and the third had a very unsuccessful racing career.”
Comment 6: L199-202: can be considered description of the study conditions, it would belong to the ‘Materials and method’ section
Response: Thank you for pointing this out. In fact, since the purpose of this information is to interpret the seasonal patterns, we have rather incorporated it in the Discussion, where it was already partly addressed (lines 252 - 255).
Comment 7: L283-287: Please document and mention studies establishing increased severity of neurological manifestations of WN fever in older horses compare to younger ones (also relative to immunity). You results showing a different pattern, it would worth a more in-depth look
Response: Thank you for requesting more references regarding this section. Authors tend to differ regarding this matter, but generally agree that younger horses are more susceptible.
“A significant, consistent decrease both in absolute WNV case numbers and in odds of WNV infection relative to other causes of neurological and febrile disease, was seen with increasing age (p=0.041), possibly due to an increased immunological resistance resulting from repeated low-grade exposure to WNV [5, 12, 13]. This is in contrast to human WNV cases in whom clinical signs are more apparent and severe in very young and very old patients, with neurological signs especially in the elderly [7, 43, 57]. Literature differs in opinion whether the age of horses influences the case fatality rate or whether geriatric horses are more or less susceptible [13, 58] but generally agree that horses younger than 5 years are more susceptible to clinical disease.” (Refer lines 307 to 314)
Comment 8: L297-299: citation needed
Response: Thank you for pointing out the oversight, the correct reference has been included here [refer line 316, reference 43)
Comment 9: L302: please insert the value of p in brackets after ‘statistically significant’ here and in similar instances (L308, L313 and so on)
Response: Thank you for indicating the need for adding the p value in the discussion text, it has been corrected as suggested. Please refer to lines 232, 234, 302, 309, 331, 338, 343.
Comment 10: L342: It might be interesting to state one out of how many pharmaceutical companies selling the WNV vaccine in the RSA?
Response: Thank you for pointing out the discrepancy, it has been corrected. There are two companies which sell WNV vaccine in RSA for horses, Zoetis and Boehringer-Ingelheim (refer line 373). Both these companies and vaccines have already been mentioned in the introduction (refer lines 132 – 135).
Comment 11: L342-346: According to the experience of most European countries, the demand for the WNV vaccine increases in those years with increased prevalence of the clinical disease (i.e. 2018 in the closest past). Then, because of the high numbers of asymptomatic individuals that gain long term acquired natural immunity and also the contribution of vaccination protocols, the following years record drastically less WN fever outbreaks, mainly until the increase of the numbers of naïve horses (new generations). In these ‘silent’ years horse owners’ awareness for vaccination change which usually leads to marked fluctuations of the WNV vaccine market, posing possible challenges for the vaccine producers. It might be of interest for the international reader community to have a South African opinion about this.
Response: Thank you for noting this. The situation in RSA is much the same for vaccinations which are not enforced by government or sport horse societies (such as African horse sickness and Equine influenza vaccinations), where owners will vaccinate prodigiously during outbreaks, and neglect to do so during “quiet” years, resulting in cyclical outbreaks of naïve populations as described above. We especially see this problem with intermittent vaccination against equine herpes virus at studs where this has been a recurrent problem (refer lines 379 to 386).
“Due to economic constraints, however, or perhaps ignorance regarding immunology, one may expect many South African horse owners to neglect regular vaccination during years when few cases occur, as is the case with other non-government regulated vaccines. This may contribute to a decline in immunity, especially in younger, immunologically naïve horses, leading to increased WNV case numbers during outbreaks. There is also uncertainty amongst horse owners regarding the duration of natural immunity post-infection, creating differing opinions regarding whether horses should be vaccinated.”
Comment 12: L345-346: please consider to rephrase ‘should have a positive effect on the numbers of WNV positive cases in horses in South Africa.’ (avoid repetition of ‘positive’; a positive effect on the positive cases involuntarily suggests an increase, which is not the case here; an addition on mentioning immunity development through both symptomatic and symptomless natural disease…)
Response: Thank you for indicating the apparent contradiction, it has been corrected by changing the sentence to “should result in decreased numbers of WNV-positive cases” (refer line 379)
Comment 13: L139: Please describe a bit more the materials and methodology of the study. For example, more details of the animals if possible (numbers of horses in each age group, use of horses, also with emphasis on their housing for possible risk factors for contracting the disease, approximate numbers of horses in the same facility for case prevalence inside an outbreak) could add benefit.
Response: Thank you for requesting this information, it has been included both in the main text and the materials and methods sections. Please refer to Figure 1 (line 196 - 197) for more information on the age groups. The use of the horses were not specifically determined during the follow-up interviews, although, in retrospect, this would have been a useful aspect to consider.
“Stabling at night, sex of horse, vaccination against African horse sickness and being generally highly stressed in the four to six weeks before clinical symptoms were detected, were not found to be significantly associated with WNV infection in the univariate analysis.”
“Additional information was obtained during telephonic interviews regarding the general vaccination status of the horses, potential stressors in the four to six weeks prior to displaying clinical signs (such as long-distance travelling, change of ownership and management system, recent illness or injury, weaning or recent African horse sickness vaccination), housing at night, outcome of the disease.
Because some of the owners had multiple submissions to the dataset (for instance at a stud), responses regarding WNV vaccination were grouped into herds based on location rather than individual horses. A total of 149 herds were interviewed (cases and controls) of which 47 herds contributed to the WNV-positive cases.
Horses were also grouped according to age, distribution of the WNV-positive cases were as follows: 56% (n=30) of cases were juveniles less than 5 years old, 39% were adult horses between the ages of 5 and 15 years old (n=21), and 5% were considered geriatric, being older than 15 years (n=3).”
(Refer to lines 350 - 352, 426 – 428).